# Antifibrotic TSG-6 Expression Is Synergistically Increased in Both Cells during Coculture of Mesenchymal Stem Cells and Macrophages via the JAK/STAT Signaling Pathway

**DOI:** 10.3390/ijms232113122

**Published:** 2022-10-28

**Authors:** Seong Chan Gong, Yongdae Yoon, Pil Young Jung, Moon Young Kim, Soon Koo Baik, Hoon Ryu, Young Woo Eom

**Affiliations:** 1Department of Surgery, Yonsei University Wonju College of Medicine, Wonju 26426, Korea; 2Regeneration Medicine Research Center, Yonsei University Wonju College of Medicine, Wonju 26426, Korea; 3Department of Internal Medicine, Yonsei University Wonju College of Medicine, Wonju 26426, Korea

**Keywords:** mesenchymal stem cells, TSG-6, hepatic stellate cells, macrophages, anti-fibrosis

## Abstract

The pro-inflammatory cytokines tumor necrosis factor-alpha (TNF-α) and interleukin (IL)-1β upregulate TNF-α-stimulated gene 6 (TSG-6); however, current knowledge about the optimal conditions for TSG-6 expression in mesenchymal stem cells (MSCs) is limited. Here, we investigated whether TSG-6 expression varies depending on the polarization state of macrophages co-cultured with adipose tissue-derived stem cells (ASCs) and analyzed the optimal conditions for TSG-6 expression in ASCs. TSG-6 expression increased in ASCs co-cultured with M0, M1, and M2 macrophages indirectly; among them, M1 macrophages resulted in the highest increase in TSG-6 expression in ASCs. TSG-6 expression in ASCs dramatically increased by combination (but not single) treatment of TNF-α, IL-1β, interferon-gamma (IFN-γ), and lipopolysaccharide (LPS). In addition, phosphorylation of signal transducer and activator of transcription (STAT) 1/3 was observed in response to IFN-γ and LPS treatment but not TNF-α and/or IL-1β. STAT1/3 activation synergistically increased TNF-α/IL-1β-dependent TSG-6 expression, and JAK inhibitors suppressed TSG-6 expression both in ASCs and macrophages. In LX-2 hepatic stellate cells, TSG-6 inhibited TGF-β-induced Smad3 phosphorylation, resulting in decreased α-smooth muscle actin (SMA) expression. Moreover, fibrotic activities of LX-2 cells induced by TGF-β were dramatically decreased after indirect co-culture with ASCs and M1 macrophages. These results suggest that a comprehensive inflammatory microenvironment may play an important role in determining the therapeutic properties of ASCs by increasing TSG-6 expression through STAT1/3 activation.

## 1. Introduction

Mesenchymal stem cells (MSCs) have been applied to treat various incurable diseases, and a substantial body of research has explored the paracrine and ant-inflammatory effects associated with their therapeutic properties. MSCs are known to migrate to damaged tissues or sites of inflammation [1,2], where they are stimulated by inflammatory cytokines such as interferon (IFN)-γ, TNF-α, or interleukin (IL)-1β to express various immunosuppressive factors, including indoleamine 2,3-dioxygenase (IDO) and prostaglandin E2 (PGE2), tumor necrosis factor (TNF)-α stimulated gene-6 (TSG-6), nitric oxide (NO), IL-6, IL-10, IL-1 receptor antagonist (IL-1ra), and human leukocyte antigen G [3,4,5,6]. Macrophages secrete various inflammatory cytokines including TNF-α and IL-1β at the site of inflammation, and they can induce the expression of TSG-6 by MSCs. When MSCs are injected intravenously, they typically form embolisms in the lungs, but they improve myocardial infarction by increasing the expression of anti-inflammatory TSG-6 [7]. In addition, intravenous administration of MSCs and recombinant TSG-6 (rTSG-6) decrease neutrophil infiltration, production of pro-inflammatory cytokines, and development of corneal opacity [8,9].

TSG-6 is a multifunctional protein associated with inflammation and exhibits anti-inflammatory and tissue-protective properties [10]. TSG-6 can be expressed in a variety of cells, including fibroblasts and MSCs, by stimulation with prostaglandin E2 (PGE2), growth factors [e.g., transforming growth factor-β (TGF-β), epidermal growth factor (EGF), and fibroblast growth factor (FGF)], and inflammatory cytokines [e.g., TNF-α, interleukin (IL)-1β] [11,12,13,14,15,16,17]. TSG-6 interacts with various extracellular matrix proteins (ECMs), such as fibronectin, heparin, heparan sulfate, hyaluronan, pentraxin-3, and thrombospondins-1 and -2, to stabilize or remodel the ECM [18,19]. In addition, TSG-6 plays an important role in inflammatory cell migration, cell proliferation, and developmental processes [20]. The anti-inflammatory properties of TSG-6 have been confirmed in experimental models of arthritis, corneal wound, myocardial infarction, colitis, and acute lung or liver injury [7,8,21,22,23,24]. TSG-6 can inhibit inflammatory responses by regulating the migration and proliferation of endothelial cells, neutrophils, mast cells, macrophage foam cells, vascular smooth muscle cells, and macrophages [25,26]. In macrophages, TSG-6 reduces inflammation by decreasing nuclear factor kappa beta translocation into the nuclei of resident macrophages in mice with zymosan-induced peritoneal inflammation [27] and acts as a negative regulator of inflammation in RAW 264.7 murine macrophage cells by producing cyclooxygenase-2 (COX-2)/prostaglandins, particularly prostaglandin D2 [28]. In addition, TSG-6 decreases the expression of p38, p-p38, c-Jun N-terminal kinase (JNK), and p-JNK and suppresses proliferation and inflammatory cytokine expression of macrophages [25]. TSG-6 reduces M1 macrophages and induces M1 to M2 transition of macrophages [29,30].

As a multifunctional protein, TSG-6 can regulate the growth of fibroblasts by suppressing the TGF-β signaling pathway, leading to antifibrotic outcomes [31,32,33]. rTSG-6 significantly decreases the viability and proliferation of capsule fibroblasts by suppressing the TGF-β/Smad2 signaling pathway in frozen shoulder and enhances cellular apoptosis concurrent with a reduction in Bcl-2 expression [31]. TSG-6 inhibits the proliferation of keloid fibroblasts by blocking the formation of the Smad2/3/4 complex and its nuclear translocation [33]. In addition, TSG-6 contributes to liver regeneration by suppressing the activation of hepatic stellate cells in CCl_4_-treated mice, suggesting that TSG-8 may have therapeutic potential in acute liver failure [23].

Taken together, MSCs exposed to an inflammatory milieu can regenerate damaged tissues by expressing TSG-6 to regulate the activity of inflammatory cells. Although inflammatory cytokines, including TNF-α and IL-1β, can increase TSG-6 expression, little is known about the relationship of TSG-6 expression between MSCs and macrophages. We hypothesized that there may be a difference in the expression of TSG-6 in MSCs associated with macrophages at different transition states. If this is the case, it is expected that the optimal timing of MSC transplantation in regenerative medicine can be determined based on exposure to inflammatory factors or macrophage transition status. Therefore, in this study, we want to identify which transition status of macrophages may be involved in TSG-6 expression in ASCs, which types of cytokines are important for TSG-6 expression, and which signaling pathways can regulate TSG-6 expression. To explore these issues, we investigated the expression of TSG-6 in both ASCs and M0, M1, and M2 macrophages after indirectly co-culturing them and the optimal condition of TSG-6 expression in ASCs. In addition, focusing on the anti-fibrotic effects of TSG-6, we investigated whether ASCs and macrophages expressing TSG-6 were able to alleviate TGF-β-induced fibrosis of hepatic stellate cells.

## 2. Results

### 2.1. Increased Expression of TSG-6 mRNA in ASCs after Coculture with Macrophages

Since MSCs induce TSG-6 expression in an inflammatory milieu, we investigated whether TSG-6 expression in MSCs differs according to macrophage transition status. ASCs were mono- or cocultured with macrophages and treated with IFN-γ and LPS, or IL-4 and IL-13, which promote macrophage transition to M1 or M2 phenotypes, respectively. *TSG-6* expression was analyzed by real-time quantitative polymerase chain reaction (qPCR). Macrophages, regardless of their transition status, increased *TSG-6* expression in ASCs. M0 and M2 macrophages increased *TSG-6* expression by 5- and 15-fold, respectively, whereas M1 macrophages significantly increased *TSG-6* expression in ASCs by approximately 62-fold (Figure 1). These results suggest that coculture with macrophages increases the expression of TSG-6 in ASCs and that M1 macrophages or pro-inflammatory conditions may play a particularly important role in TSG-6 expression in ASCs.

Next, we analyzed whether macrophages were essential for TSG-6 expression in ASCs and whether TSG-6 was expressed in macrophages. ASCs and macrophages were mono- or cocultured and treated with IFN-γ and LPS (macrophage transition factors promoting the M1 phenotype). In ASCs treated with IFN-γ and LPS, *TSG-6* expression was increased by approximately 21-fold, while ASCs cocultured with M0 macrophages exhibited an approximately 10-fold increase in *TSG-6* expression. Interestingly, when ASCs were cocultured with M1 macrophages (differentiated with IFN-γ and LPS), *TSG-6* mRNA increased approximately 63- and 46-fold in ASCs and M1 macrophages, respectively (Figure 2A black bars). Similar to the pattern of *TSG-6* mRNA expression shown in Figure 2A, TSG-6 expression significantly increased in both cell types when M1 macrophage and ASCs were cocultured (Figure 2B). Since a marked upregulation of *TSG-6* mRNA was observed in phorbol ester-induced differentiation of THP-1 monocytes to macrophages [34], we analyzed whether macrophages specifically were necessary for TSG-6 expression in ASCs using murine macrophage cells Raw 264.7 to eliminate the use of phorbol esters for transdifferentiation. Similar to M1 macrophages differentiated from THP-1, Raw264.7 cells treated with LPS significantly induced TSG-6 expression in ASCs (Figure 2C). These results suggest that both macrophages and MSCs can overexpress TSG-6 if they coexist in an inflammatory or injured site.

### 2.2. TSG-6 Expression in ASCs and Macrophages Treated with Cytokine Combinations

Since TNF-α and IL-1β secreted by macrophages can induce TSG-6 expression, we investigated whether IFN-γ and LPS, TNF-α, or IL-1β play a role in the increase of TSG-6 expression in ASCs. In ASCs treated individually with IFN-γ and LPS, TNF-α, or IL-1β, elevated TSG-6 levels were only detected in immunoblots exposed for a long period (300 s). However, in ASCs treated with a combination of IFN-γ, LPS, and TNF-α, a significant increase in TSG-6 expression was observed even in immunoblots exposed for short periods (22 s), with this increase dependent on the doses of TNF-α (Figure 3A, lanes 5 and 6) but not IL-1β (Figure 3A, lanes 7 and 8). However, maximum TSG-6 expression was observed when cells were treated with all four factors (Figure 3A, lanes 9 and 10). These results suggest that the synergistic action of several inflammatory cytokines is required for optimal expression of TSG-6 in ASCs. In contrast, increased TSG-6 expression in macrophages was observed in response to combinatorial treatment with IFN-γ, LPS, and TNF-α, compared to cells treated with IFN-γ and LPS (Figure 3B, lane 5). The expression pattern of TSG-6 secreted into the culture supernatant was quite similar to that observed in the cell lysate (Figure 3). These results suggest that the conditions for inducing TSG-6 expression differ according to cell types, and that ASCs can express TSG-6 by responding more sensitively to inflammatory conditions than do macrophages.

### 2.3. Activation of the Signal Transducer and Activator of Transcription (STAT) Signaling Pathway for TSG-6 Expression

Next, we investigated whether the STAT pathway, which is activated by IFN-γ and LPS, is involved in TSG-6 expression.

In both ASCs and macrophages, IFN-γ + LPS treatment, but not TNF-α or IL-1β treatment, induced STAT1/3 phosphorylation at 15 min (Figure 4). The Janus kinase (JAK) inhibitor that inhibits STAT activity reduced TSG-6 expression induced by the cytokine combination treatment (Figure 5A). In addition, this decrease in TSG-6 expression was also observed in both ASCs and macrophages cocultured indirectly (Figure 5B,C). These results suggest that the STAT signaling pathway induced TSG-6 expression in ASCs and macrophages. Furthermore, STAT activity induced a synergistic expression of TSG-6 during TNF-α + IL-1β treatment. Taken together, IFN-γ + LPS, TNF-α, or IL-1β can induce TSG-6 expression in ASCs and macrophages, and when ASCs and macrophages are exposed to TNF-α or IL-1β under STAT-activated conditions, TSG-6 overexpression may occur.

### 2.4. Antifibrotic Effects of TSG-6 Expressed in ASCs and Macrophages

In addition to its representative anti-inflammatory properties, TSG-6 may modulate fibrosis by regulating the TGF-β/Smad pathway [31,33]. Therefore, we investigated whether TSG-6 expressed by ASCs and macrophages could regulate the fibrotic process in LX-2 cells induced by TGF-β. In LX-2 cells, TGF-β induced Smad3 phosphorylation and α-SMA expression, which were significantly decreased by TSG-6 (Figure 6A,B). In addition, α-SMA expression induced by TGF-β was significantly reduced in LX-2 cells cocultured with ASCs and macrophages indirectly. We used phorbol ester 12-O-tetradecanoylphorbol-13-acetate (TPA) to differentiate THP-1 monocytes into macrophages; however, TPA is an activator of protein kinase C (PKC) that can regulate the TGF/Smad pathway [35,36]. Therefore, to confirm the antifibrotic effects of TSG-6 alone, excluding the effect of TPA in the coculture of ASCs and macrophages, Raw 264.7 macrophages were used. As shown in Figure 6D, ASCs and Raw 264.7 cells induced a decrease in α-SMA expression in LX-2 cells compared to TGF-β-treated cells.

Next, to determine whether the reduction of α-SMA expression in LX-2 was dependent on TSG-6 expressed by ASCs and macrophages, we investigated the expression of α-SMA after *TSG-6* silencing using siRNA. ASCs and Raw 264.7 macrophages in the upper chamber were treated with scrambled or *TSG-6*-specific siRNA and then cocultured with LX-2 cells. TSG-6 expression was significantly reduced in *TSG-6* siRNA-treated ASCs and Raw 264.7 macrophages (Figure 7A), which resulted in the recovery of α-SMA expression in TGF-β-exposed LX-2 cells (Figure 7B).

These results suggest that TSG-6 can decrease α-SMA expression by inhibiting Smad3 phosphorylation in LX-2 cells, and TSG-6 expressed by ASCs and macrophages regulates the fibrotic activity of LX-2 cells.

## 3. Discussion

Although MSCs can express TSG-6, we found that M1 macrophages induced the highest increase in TSG-6 expression in ASCs. In addition, combination treatments of IFN-γ + LPS, TNF-α, and IL-1β were responsible for a significant increase in TSG-6 expression in ASCs via phosphorylation of STAT1/3. STAT1/3 phosphorylation was observed in response to treatment with IFN-γ and LPS, but not TNF-α and/or IL-1β. STAT1/3 activation synergistically increased TNF-α/IL-1β-dependent TSG-6 expression, and JAK inhibitors suppressed TSG-6 expression in both ASCs and macrophages. TSG-6 inhibited TGF-β-induced Smad3 phosphorylation and resulted in decreased α-SMA expression. Moreover, fibrotic activities of LX-2 cells induced by TGF-β were dramatically decreased after indirect coculture with ASCs and M1 macrophages.

Although inflammatory mediators (e.g., IL-1β, TNF-α, LPS, TGF-β, and PGE2) have been shown to induce TSG-6 expression in leukocytes, stromal cells, and several tissues, optimal conditions of TSG-6 expression in each cell type have not been accurately identified [10,18,37,38]. In this study, a significant increase in TSG-6 expression was observed in both cocultured ASCs and macrophages. Macrophages secrete various inflammatory cytokines including TNF-α and IL-1β at the site of inflammation, and they can induce the expression of TSG-6 by ASCs. In addition, a significant increase in TSG-6 expression was observed in cocultured ASCs treated with the combination of IFN-γ, LPS, and TNF-α/IL-1β, but not in macrophages cultured alone. These results indicate that, although the stimuli that induce TSG-6 expression in ASCs and macrophages are different, the co-localization of ASCs and macrophages is important to express TSG-6 in an inflammatory or injured site. In macrophages, stimulation other than the combination of IFN-γ, LPS, and TNF-α is required for optimal TSG-6 expression. PGE2 induces M1-to-M2 transition of macrophages [39,40] and increases TSG-6 expression [11]. PGE2 production is increased in both cells during coculture of MSCs and macrophages [40,41]. In addition to PGE2, kynurenic acid (KYNA) can increase TSG-6 expression in monocytes [42]. KYNA is an indoleamine 2,3-dioxygenase (IDO) metabolite, which is produced during tryptophan catabolism. MSCs exposed to IFN-γ and TNF-α express IDO, which can have anti-inflammatory and immunosuppressive functions [43]. Taken together, these data indicate that different cells express TSG-6 at different intensities depending on the treatment combination of inflammatory mediators (e.g., IL-1β, TNF-α, IFN-γ, LPS, TGF-β, PGE2, and KYNA). Therefore, when MSCs are applied to regenerative medicine, the optimal therapeutic effects can be expected by transplanting MSCs in consideration of the patient’s inflammatory cytokine profile or the transition state of macrophages. More than 50 cytokines, growth factors, and hormones play important roles in development, metabolism, and healing by activating STAT-mediated signaling [44,45,46,47]. In this study, unlike TNF-α or IL-1β, IFN-γ and LPS induced M1 macrophage differentiation of TPA-treated monocytes, and phosphorylation of STAT1/3 in ASCs and macrophages. Inhibitors of JAKs (signaling molecules upstream of STAT) significantly reduced TSG-6 expression in ASCs and macrophages treated with cytokine combinations or cocultured. These results suggest that the STAT-mediated signaling pathway may play a critical role in directly increasing TSG-6 expression or regulating STAT-mediated IDO or COX-2 expression. To verify this, it will be necessary to assess the changes in TSG-6 expression in the presence of IDO and COX-2 inhibitors. Therefore, we plan further studies to understand the precise mechanisms by which the STAT-mediated signaling pathways increase TSG-6 expression.

In conclusion, considering the inflammatory environment during MSC transplantations will be very important to enhance the anti-inflammatory and antifibrotic properties of TSG-6 expressed by MSCs and macrophages.

## 4. Materials and Methods

### 4.1. Materials

The reagents used in this study were obtained from the indicated suppliers: IFN-γ, IL-1β, TNF-α, IL-4, IL-13, and TGF-β from R&D Systems (Minneapolis, MN, USA); antibodies against α-SMA (ab7817) from Abcam (Cambridge, UK); antibodies against GAPDH (sc47724) and TSG-6 (sc377277) from Santa Cruz Biotechnology (Santa Cruz, CA, USA); antibodies against STAT1 (9172S), pSTAT1 (9167S), STAT3 (9139S), pSTAT3 (9145S), and Smad2/3 (8685S) from Cell Signaling Technology (Danvers, MA, USA); antibodies against pSmad2/3 (PA5-110155) from ThermoFisher Scientific (Waltham, MA, USA); chemical inhibitors for JAK1 (Abrocitinib, HY-107429; MedChemExpress, Princeton, NJ, USA) and JAK2 (AG-490, S1143; Selleck Chemicals, Houston, TX, USA); scramble and *TSG-6* siRNA (Santa Cruz). All other materials were purchased from Sigma-Aldrich (St. Louis, MO, USA) unless otherwise indicated.

### 4.2. Cell Culture

The hepatic stellate cell line LX-2 was purchased from Millipore (Burlington, MA, USA), and THP-1 monocytes and Raw 264.7 murine macrophage cells were purchased from the Korea Cell Line Bank (Seoul, Korea). LX-2 or Raw 264.7 cells were maintained in Dulbecco’s Modified Eagle Medium (DMEM, Gibco BRL, Rockville, MD, USA) supplemented with 3% or 10% fetal bovine serum (FBS, Gibco BRL), respectively. ASCs were isolated from three healthy donors (24–38 years of age) with their written informed consent through elective liposuction procedures under anesthesia at the Wonju Severance Christian Hospital (Wonju, Korea). ASCs were maintained in DMEM supplemented with 10% FBS, and cells from passages 3–5 were used in all experiments. THP-1 cells were sub-cultured with Roswell Park Memorial Institute (RPMI) 1640 Medium (Gibco BRL) supplemented with 10% FBS. Penicillin/streptomycin (Gibco BRL) was also supplemented in culture media for maintaining cells at 37 °C and 5% CO_2_. For experiments, cells were seeded for 24 h, and treated with stimuli for the indicated time. Cells were exposed to JAK inhibitors 20 min prior to stimulus treatment.

For indirect coculture, Transwell plates (SPL, Pocheon, Korea) were used. To evaluate the anti-fibrotic effects of TSG-6, LX-2 cells were seeded in the lower chamber, and ASCs and macrophages were directly seeded into the upper chamber. Briefly, THP-1 monocytes were differentiated into macrophages for two days in the upper chamber, and LX-2 cells were plated in the lower chamber a day before coculture. To proceed with the coculture of ASCs (1 × 10^4^ cells/cm^2^), macrophages (2.5 × 10^4^ cells/cm^2^), and LX-2 cells (1 × 10^4^ cells/cm^2^), the upper chamber seeded with macrophages was assembled over the lower chamber, and ASCs were immediately added to the upper chamber.

### 4.3. Macrophage Differentiation of THP-1 Monocytes

THP-1 monocytes were differentiated into macrophages (M0 phase) using 100 nM TPA (Sigma-Aldrich) for two days. M0 macrophages were washed with phosphate-buffered saline (PBS, Welgene, Gyeongsan, Korea), and M1 or M2 transition was induced by treatment with 20 ng/mL of IFN-γ and 10 pg/mL of LPS (Sigma-Aldrich) or 20 ng/mL each of IL-4 and IL-13 for two more days, respectively (Appendix A).

### 4.4. qPCR

Total RNA was extracted using TRIzol reagent (Gibco BRL) according to the manufacturer’s instructions. cDNA was synthesized from 1 μg of total RNA using Verso cDNA synthesis kit (ThermoFisher Scientific). *TSG-6* mRNA was amplified using the sense and antisense primers 5′-TGGCTTTGTGGGAAGATACTGT-3′ and 5′-TGGAAACCTCCAGCTGTCAC-3′, respectively. For *GAPDH*, sense and antisense primers of 5′-CAAGGCTGAGAACGGGAAGC-3′ and 5′-AGGGGGCAGAGATGATGACC-3′, respectively were used. The reagents in a 10-μL reaction mixture included cDNA, primer pairs, and SYBR Green PCR Master Mix (Applied Biosystems, Dublin, Ireland), and PCR was conducted using a QuantStudio 6 Flex Real-time PCR System (ThermoFisher Scientific). All qPCR reactions were performed in triplicate. *GAPDH* expression was used for normalization. The 2^−(ΔΔCt)^ method was used to calculate relative fold changes in mRNA expression.

### 4.5. Immunoblotting

Cells were lysed in sample buffer [62.5 mM Tris-HCl, pH 6.8, 34.7 mM sodium dodecyl sulfate (SDS), 10% (*v*/*v*) glycerol, and 5% (*v*/*v*) β-mercaptoethanol], boiled for 5 min, subjected to SDS-polyacrylamide gel electrophoresis, and transferred to an Immobilon membrane (Millipore). After blocking with 5% skim milk in Tris-HCl-buffered saline containing 0.05% (*v*/*v*) Tween 20 (TBST) for 30 min, the membrane was incubated with primary antibodies against TSG-6, α-SMA, pSmad2/3, and GAPDH at a dilution of 1:1000 or STAT1, pSTAT1, STAT3, pSTAT3, and Smad2/3 at a dilution of 1:2000 at 4 °C overnight. The membrane was washed thrice for 5 min with TBST and then incubated with horseradish peroxidase-conjugated secondary antibodies (1:5000; 7074S and 7076S, Cell Signaling Technology) for 1 h. After washing thrice with TBST, protein bands were visualized using an EZ-Western Lumi Pico or Femto kit (Dogen, Seoul, Korea) and detected using a ChemiDoc XRS+ system (Bio-Rad, Hercules, CA, USA). To detect TSG-6 expression, the membrane made from ASCs treated with TNF-α or IL-1β was long exposed (approximately 300 s), whereas the membrane obtained from the ASCs after IFN-γ, LPS, and TNF-α treatment was sufficient with a short exposure (approximately 22 s). The intensity of immunoreactive bands was quantified by densitometry using ImageJ, and relative expression of proteins was normalized with respect to GAPDH expression.

### 4.6. Small Interfering RNA (siRNA) Treatment

To inhibit TSG-6 expression in ASCs and macrophages, ASCs/macrophages or LX-2 cells were seeded in the upper or lower chamber of Transwell wells, respectively. *TSG-6* siRNA (Santa Cruz) or scrambled siRNA (Bioneer, Daejeon, Korea) was mixed with Lipofectamine RNAiMAX (Invitrogen, Carlsbad, CA, USA) according to the manufacturer’s recommendations and applied to the cells, which were incubated for two days.

### 4.7. Statistical Analyses

All experiments were performed thrice. Data are expressed as the mean ± standard deviation. *p* values were determined using a paired 2-tailed Student’s *t*-test (Mann–Whitney U test). All statistical analyses were performed using GraphPad Prism 7.0 software (GraphPad Inc., La Jolla, CA, USA). Significance was set at *p* ≤ 0.05.

## Figures and Tables

**Figure 1 ijms-23-13122-f001:**
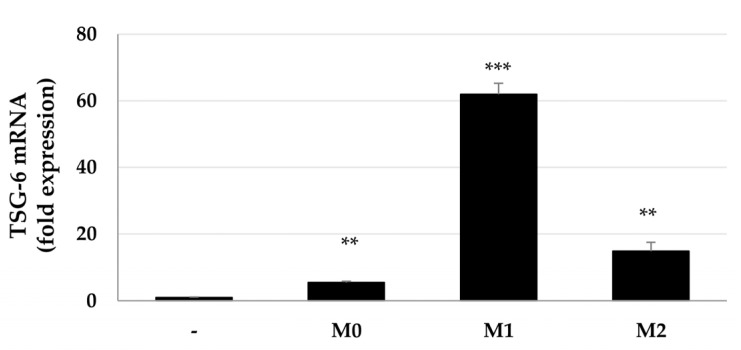
*TSG-6* mRNA expression in ASCs indirectly cocultured with macrophages. ASCs were indirectly cocultured with macrophages for 2 days and *TSG-6* mRNA expression assessed by qPCR. All qPCR reactions were performed in triplicate. *GAPDH* expression was used for normalization. The 2^−(ΔΔCt)^ method was used to calculate relative fold changes in mRNA expression. Data are presented as the mean ± standard deviation (SD) of three independent experiments. ** *p* ≤ 0.01 and *** *p* < 0.001. M0, macrophages differentiated from THP-1 monocytes; M1, M0 macrophages treated with IFN-γ and LPS; M2, M0 macrophages treated with IL-4 and IL-13.

**Figure 2 ijms-23-13122-f002:**
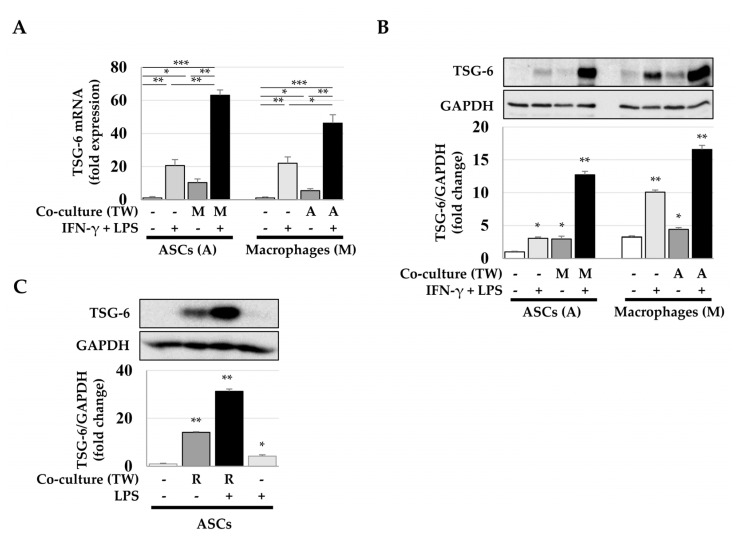
TSG-6 expression in ASCs and macrophages after indirect coculture. For indirect coculture using Transwell plates, ASCs and macrophages were seeded in upper and lower chamber (and vice versa) and then total RNA and protein samples were prepared from cells in the lower chamber. (**A**) Relative *TSG-6* mRNA expression in ASCs and macrophages after indirect coculture. (**B**) Relative TSG-6 expression in ASCs and macrophages after indirect coculture. (**C**) TSG-6 expression in ASCs cocultured with RAW 264.7 murine macrophages. ASCs and RAW 264.7 cells were seeded in lower and upper chamber, respectively, and the TSG-6 expression in ASCs was analyzed using immunoblotting. Relative expression was normalized with respect to GAPDH expression. Data are presented as the mean ± SD of three independent experiments. * *p* ≤ 0.05, ** *p* ≤ 0.01, and *** *p* < 0.001. TW, Transwell; A, ASCs; M, macrophages; R, RAW 264.7 murine macrophages.

**Figure 3 ijms-23-13122-f003:**
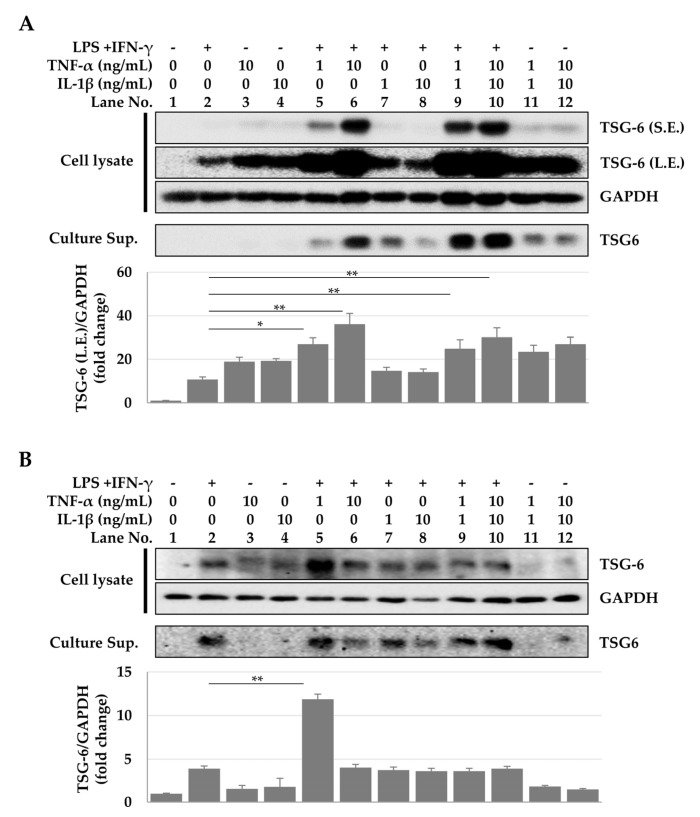
TSG-6 expression in ASCs and macrophages treated with TNF-α, IL-1β, and/or IFN-γ + LPS. ASCs or macrophages were treated with different concentration of TNF-α, IL-1β, and/or IFN-γ (20 ng/mL) + LPS (10pg/mL) for 48 h, and TSG-6 expression was detected in the cell lysate and culture supernatant using immunoblotting. (**A**) TSG-6 expression in ASCs. (**B**) TSG-6 expression in macrophages. The data shown represent one of three independent experiments. * *p* ≤ 0.05 and ** *p* ≤ 0.01. S.E., short exposure (22 s); L.E. long exposure (300 s); Sup., supernatant.

**Figure 4 ijms-23-13122-f004:**
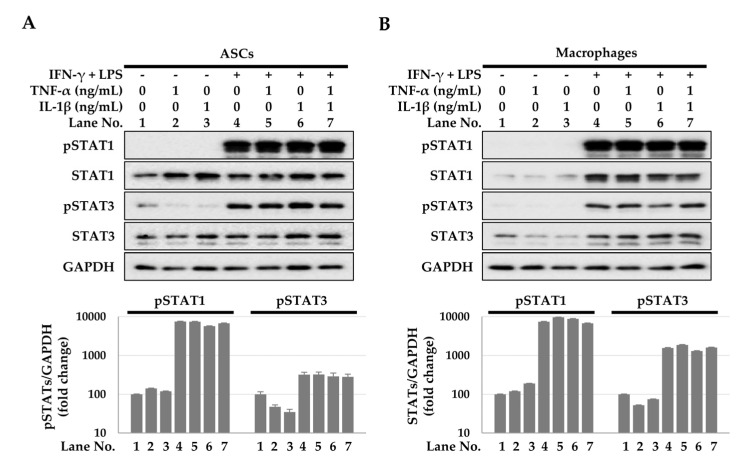
STAT1/3 phosphorylation in ASCs and macrophages. To detect STAT1/3 phosphorylation, protein samples were prepared after 15 min of treatment with TNF-α, IL-1β, and/or IFN-γ + LPS and analyzed using immunoblotting in ASCs (**A**) and macrophages (**B**). The data shown represent one of three independent experiments.

**Figure 5 ijms-23-13122-f005:**
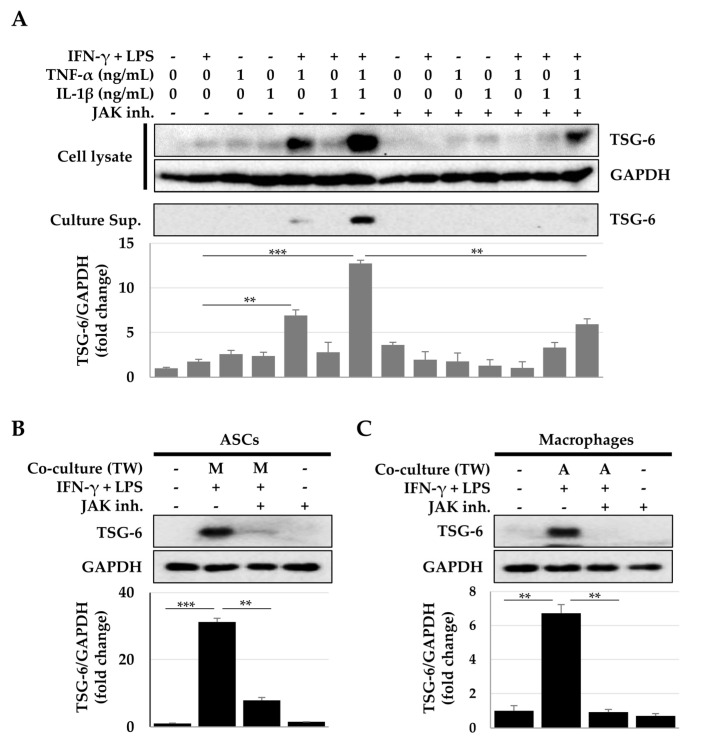
Inhibition of TSG-6 expression by JAK inhibitors. Cells were treated with JAK inhibitors (10 µM each of JAK1 and JAK2 inhibitor) 20 min prior to cytokine treatment or coculture. Cell lysates were recovered 48 h after stimulation, and TSG-6 expression was analyzed using immunoblotting. (**A**) TSG-6 expression in ASCs after treatment with TNF-α, IL-1β, and/or IFN-γ + LPS. TSG-6 expression in ASCs (**B**) or macrophages (**C**) after coculture. The data shown represent one of three independent experiments. ** *p* ≤ 0.01 and *** *p* < 0.001. Inh., inhibitor.

**Figure 6 ijms-23-13122-f006:**
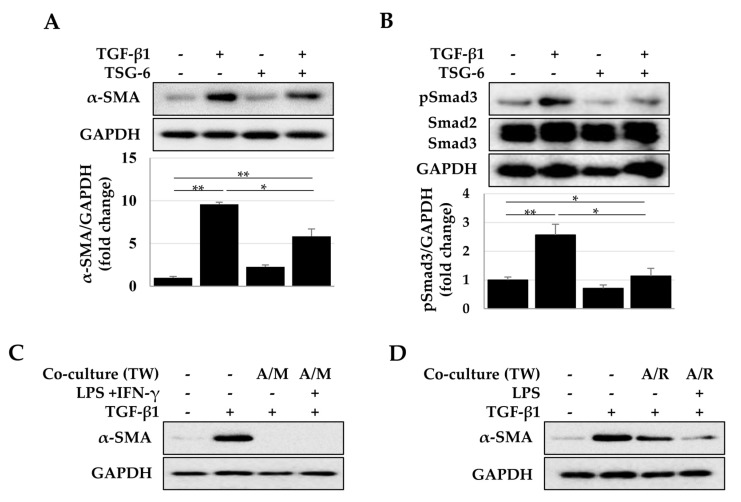
Antifibrotic effects of TSG-6 in TGF-β-treated LX-2 cells. (**A**) Inhibition of α-SMA expression by TSG-6 in TGF-β-treated LX-2 cells. (**B**) Inhibition of Smad3 phosphorylation by TSG-6. Expression of α-SMA (**A**) and pSmad3 (**B**) was quantified by densitometry using Image J, and relative expression was normalized with respect to GAPDH expression. Data are presented as the mean ± SD of three independent experiments. * *p* ≤ 0.05 and ** *p* ≤ 0.01. LX-2 cells in the lower chamber were cocultured with ASCs and macrophages seeded in the upper chamber. Inhibition of α-SMA expression in LX-2 cells cocultured with ASCs and macrophages (**C**) or ASCs and RAW 264.7 macrophages (**D**). TW, Transwell.

**Figure 7 ijms-23-13122-f007:**
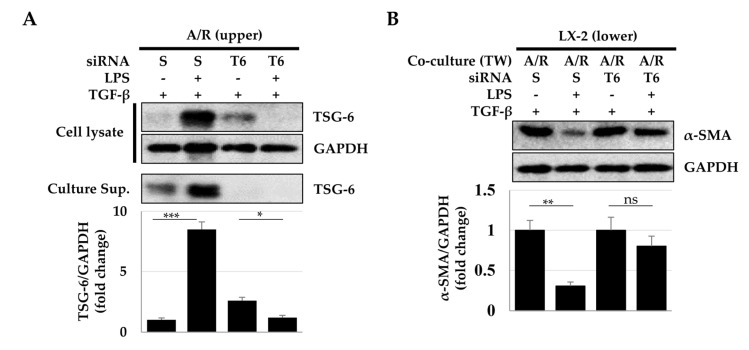
Effects of *TSG-6* siRNA treatment in ASCs and macrophages. To inhibit TSG-6 expression in ASCs and macrophages, ASCs/macrophages or LX-2 cells were seeded in the upper or lower chamber of Transwell wells, respectively. (**A**) TSG-6 expression in ASCs and macrophages 48 h after transfection of *TSG-6* siRNA. (**B**) α-SMA expression in TGF-β-treated LX-2 cells 48 h after transfection of *TSG-6* siRNA. Data were obtained from one of three independent experiments. * *p* ≤ 0.05, ** *p* ≤ 0.01, and *** *p* < 0.001. ns, no statistical significance.

## Data Availability

The data that support the findings of this study are available from the corresponding author upon reasonable request.

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
