# Peer review of "Antifibrotic TSG-6 Expression Is Synergistically Increased in Both Cells during Coculture of Mesenchymal Stem Cells and Macrophages via the JAK/STAT Signaling Pathway"

_ijms, 2022, doi:10.3390/ijms232113122_

Round 1
Reviewer 1 Report
This work is devoted to the investigation of the relation between the expression of TNF-α-stimulated gene 6 (TSG-6) and polarization state of macrophages co-cultured with adipose tissue-derived stem cells (ASCs). The signalling pathway involved in increased expression of antifibrotic TSG-6 was also investigated. Using a variety of methods the authors found that the expression TSG-6 was significantly increased in ASCs co-culture with M0, M1, and M2 macrophages, the highest level being obtained with M1 macrophages. Signal transducer and activator of transcription (STAT) 1/3 increased TSG-6 expression TNF-α/IL-1β dependent, while JAK inhibitors suppressed gene expression in both cell lines used as experimental model. Also, α-smooth muscle actin (fibrotic marker) expression induced by TGF-β in hepatic stellate cells LX-2 was decreased after co-culture with ASCs and M1 macrophages due to the inhibition of TGF-β-induced Smad3 phosphorylation.
The results are important to define the optimal conditions in order to increase the expression of TSG-6 in MSCs and macrophages and thus to enhance its anti-inflammatory and antifibrotic properties.
The subject of the manuscript addressed by the authors is interesting and comprehensive presented.
Based on my comments the paper can be published in the present form.
Author Response
We are grateful for the reviewer’s appreciation of our work. To address the opinion that a spell check is necessary, we thoroughly proofread the revised manuscript for any spelling or grammatical errors.
Reviewer 2 Report
In this work entitled “Antifibrotic TSG-6 expression is synergistically increased in both cells during co-culture of mesenchymal stem cells and macrophages via the JAK/STAT signaling pathway” the authors aimed to comprehend how the inflammatory microenviroment modulates TSG-6 expression in ASCs and macrophages. However, the novelty and relevance of this work is not clear. The authors should improve the whole manuscript. It is not well organized and the figures are poor and not self-explanatories. Several concerns must be addressed for its publication:
Introduction.
Authors must clarify the aim of the present work. It is not clear the connection between TSG-6, MSCs and macrophages. Please, improve the introduction (it is very confused without a conductive thread) and remark the aim and hypothesis of the work.
Results.
Lines 93-95. Please, clarify the effect of the co-culture on TSG-6 expression.
Lines 95-97. “These results suggest that M1 macrophages or pro-inflammatory conditions are very important for TSG-6 expression in ASCs” Please, improve this sentence, the language is not appropriate.
Lines 105-107. “macrophages were 105 mono- or cocultured and treated with IFN-γ and LPS that are macrophage transition fac-106 tors to M1 phenotype” This information must appear previously. It is necessary to understand figure 1.
Lines 107-109. Please, improve the sentence. It is not clear which is the treatment in each condition (ASCs + IFNg and LPS or ASCs cocultured with Mo and treated with IFNg and LPS).
Lines 109-111. “Interestingly, approximately 63-fold increased expression of TSG-6 mRNA was observed in ASCs cocultured with M1 macrophages that were differentiated with IFN-γ and LPS” Please, amend (avoid being repetitive).
Lines 135-137. “To investigate the synergistic TSG-6 expression conditions, ASCs and macrophages were treated with the combination of IFN-γ and LPS that are used for M1 transition of macrophages, TNF-α, and IL-1β, which induce TSG-6 expression” Please, amend (avoid being repetitive).
Figure 3. Please, add numbers to the different lines of the WB and refer to this numbers in the results. The blot has a lot of conditions, it is difficult to compare the different treatments. Another option is to incorporate a color code or a scheme. This figure must be improved.
Additionally, in this figure the authors incorporate “culture sup”. However, they did not mention this condition in the result section and it is not explain in the legend of the figure.
The legend also must be improved. Please, add the concentration of LPS and IFNy used. Further, authors have to clarify in the text why they included the results of long and short exposures and describe and justify the results.
Is it a representative blot? Authors must mention the number of replicates performed. This reviewer also suggest to include a densitometric analysis for a better comprehension and for statistical analysis.
Figure 4 and 5. Legend must be improved. Please, incorporate information about the replicates and a densitometric analysis. In figure 5 information about JAK inhibitors must be added.
Discussion.
The discussion is not well organized, it must be improved. This reviewer suggest that the first paragraph of the discussion summarizes the relevance of the present work. Additionally, it would be better to incorporate more interpretation of the results. In the actual version, it is like an enumeration of antecedents.
Materials and methods.
Cell culture. The number of cells used in each experiment must be incorporate in this section.
Immunoblotting. Authors should mentioned the catalogue number of the antibodies used for WB.
Author Response
We are grateful for the reviewer’s insightful and constructive critique of our work. Below, we provide our responses to the reviewer’s questions and describe how we have addressed them to improve our manuscript.

Round 2
Reviewer 2 Report
Although the authors improved the manuscript, there are still important things that need to be resolved.
- The figures could be improved. The legends are poor, the reader needs to go to the text to fully understand the figure.
- Densitometric analysis of WB was not included.
- The information about exposure in WB must be included in the MyM section.
- Authors must clarify which JAK inhibitors were used (type of molecule, supplier, concentration, etc)
Additionally, did authors prove that macrophages differentiation was ok? Did they measure specific markers of M1 and M2?
Author Response
We are grateful for the reviewer’s insightful and constructive critique of our work. We provide our responses to the reviewers’ comments and describe how we have addressed them to improve our manuscript.

Round 3
Reviewer 2 Report
Although it is difficult to establish whether this work has a significant relevance in the field, the manuscript can be published in its current version.